



# Assessment of negative and positive CO₂ emissions on global warming metrics using large ensemble Earth system model simulations

Negar Vakilifard[1], Richard G. Williams[2], Philip B. Holden[1], Katherine Turner[2,3], Neil R. Edwards[1,4], and David J. Beerling[5]

[1]Environment, Earth and Ecosystems, The Open University, Milton Keynes, UK

[2]Department of Earth, Ocean and Ecological Sciences, School of Environmental Sciences, University of

Liverpool, Liverpool, United Kingdom

[3]Leverhulme Research Centre for Functional Materials Design, Liverpool, United Kingdom

[4]Cambridge Centre for Energy, Environment and Natural Resource Governance, University of Cambridge, Cambridge, UK

[5]Leverhulme Centre for Climate Change Mitigation, School of Biosciences, University of Sheffield,

Sheffield, UK

*Correspondence to*: Negar Vakilifard (negar.vakilifard@open.ac.uk)

Keywords: effective TCRE, carbon cycle feedback, climate feedback, ocean heat uptake, hysteresis of the Earth system, negative emissions, zero emissions commitment, uncertainty in the Earth system's

feedbacks



## Abstract

The benefits of implementing negative emission technologies for a century (years 2070-2170) on the global warming response to cumulative carbon emissions until year 2420 are assessed with a comprehensive set of intermediate complexity Earth system model integrations. Model integrations include 82 different model realisations covering a wide range of plausible climate states. The global warming response is assessed in terms of two key climate metrics: the effective transient climate response to cumulative $CO_2$ emissions (eTCRE), measuring the surface warming response to cumulative carbon emissions and associated non-$CO_2$ (RCP4.5) forcing, and the zero emissions commitment (ZEC), measuring the extent of any continued warming after net zero is reached. The TCRE is approximated from eTCRE by removing the contributions of non-$CO_2$ forcing as 2.15 °C EgC$^{-1}$ (with a 10-90 % range of 1.6 to 2.8 °C EgC$^{-1}$). During the net positive emission phases, the eTCRE decreases from 2.62 (1.90 to 3.65) to 2.30 (1.73 to 3.23) °C EgC$^{-1}$ due to a weakening in the increase in radiative forcing with an increase in atmospheric carbon, which is partly opposed by an increasing fraction of the radiative forcing warming the surface as the ocean stratifies. During the negative and zero emission phases, a progressive reduction of the eTCRE to 2.0 (1.4 to 2.8) °C EgC$^{-1}$ is driven by the reducing airborne fraction as $CO_2$ is drawn down by the ocean. The model uncertainty in the slopes of warming versus cumulative $CO_2$ emissions varies from being controlled by the radiative feedback parameter during positive emissions to also being affected by ocean circulation and carbon-cycle parameters during zero or net-negative emissions. There is hysteresis in atmospheric $CO_2$ and surface warming, where atmospheric $CO_2$ and surface temperature are higher after peak emissions compared with before peak emissions. The continued warming after emissions cease defining the ZEC for the model mean without carbon capture is -0.01 °C at 25 years and decreases in time to -0.15 °C at 90 years after emissions cease. However, there is a spread in the ensemble with a temperature overshoot occurring in 50 % of the ensemble members at year 25. The ZEC only becomes negative in all ensemble members if modest carbon capture is included. Hence, incorporating negative emissions enhances the ability to meet climate targets and avoid risk of continued warming after net zero is reached.



## 1 Introduction

There is an increasing need to adopt negative emission technologies (Luderer et al., 2013; Rogelj et al., 2015; Beerling et al., 2020) to enhance the chance of meeting the Paris climate agreement targets of global warming of 1.5 °C or less than 2 °C given the ongoing growth in greenhouse gas concentrations (Boucher et al. 2012; Jeltsch-Thömmes et al., 2020). For a 1.5 °C target, there is 66 % chance of meeting this target only if post-2019 cumulative carbon emissions are limited to less than ~400 $GtCO_2$ (IPCC 2021). How much carbon may be emitted while

remaining within the warming target is inversely proportional to the amount of surface warming resulting from cumulative carbon emissions. The increase in the mean-global surface air temperature relative to cumulative $CO_2$ emission is defined as the transient climate response to cumulative $CO_2$ emissions (TCRE) (Matthews et al., 2009; Zickfeld et al., 2012; IPCC, 2013; Gillett et al., 2013; Zickfeld et al., 2013; Friedlingstein et al., 2014; Matthews et al., 2017). Climate model projections reveal a simple near-linear relationship between the global surface air

temperature change and cumulative $CO_2$ emissions between 0 and ~2000 PgC (MacDougall, 2016). However, despite a similar linear dependence, there is a wide inter-model range in TCRE values (Williams et al., 2017; Spafford and MacDougall, 2020), varying from 1.4 to 2.5 °C $TtC^{-1}$ in intermediate-complexity Earth system models (Eby et al., 2013), 0.8 to 2.4 °C $TtC^{-1}$ in full-complexity Earth system models (Matthews et al., 2018), and 0.7 to 2 °C $TtC^{-1}$ (90 % confidence interval) in observationally-constrained TCRE estimates from a 15-member

CMIP5 ensemble (Gillett et al., 2013).

For the case of radiative forcing exclusively from atmospheric $CO_2$, the TCRE can be related to the dependence of the global mean temperature on the radiative forcing, the dependence of the radiative forcing on the atmospheric $CO_2$ and the airborne fraction (Sect. 2; Williams et al., 2016; Ehlert et al., 2017; Katavouta et al., 2018; Williams et al., 2020). Applying this framework to 7 CMIP5 and 9 CMIP6 models following a 1 % annual

increase in atmospheric $CO_2$, the TCRE is affected by a large inter-model spread in the climate feedback parameter for CMIP6 (Williams et al., 2020) as well as by a larger inter-model spread in the land carbon system for CMIP5 (Jones and Friedlingstein, 2020). The inclusion of non-$CO_2$ radiative forcing is able to alter the relationship between emissions and surface warming through both direct warming and carbon feedback effects (Tokarska et al., 2018). For the more realistic case including non-$CO_2$ radiative forcing contributions, the TCRE may be

estimated by approximately removing the warming linked to non-$CO_2$ radiative forcing (Matthews et al., 2021). Alternatively, an effective TCRE (eTCRE) may be defined to include non-$CO_2$ warming and the non-$CO_2$ radiative forcing (Williams et al., 2016; Williams et al., 2017).





Here we apply the eTCRE framework developed by Williams et al. (2016) to the intermediate complexity Earth system model GENIE-1. The use of intermediate complexity enables us to (i) quantify uncertainties through a large ensemble consisting of 82 members that simulate a wide range of plausible climate states and (ii) explore long time scales, in both the historical and future periods. The model was spun up to preindustrial and integrated from years 850 to 2420 CE, extending several centuries after the emissions cease. This extension of the model allows assessment of continued warming at certain points after emissions cease, referred to as the zero emissions commitment (ZEC). For our eTCRE analysis, we chose 850 CE as the preindustrial baseline (Eby et al., 2013) so we can account for both land use change and fossil fuel $CO_2$ emissions. Our simulations follow two scenarios of: (a) $CO_2$ emissions-forced RCP4.5 as a potential future medium-level mitigation scenario, in which the radiative forcing stabilises at 4.5 $Wm^{-2}$ before year 2100, by the employment of a range of greenhouse gas mitigation technologies and strategies; and (b) $CO_2$ emissions-forced RCP4.5 with additional point-source carbon capture and storage. The carbon capture and storage is applied with 50 years delay in action to allow investigating the controls of uncertainty in TCRE during both positive and net negative emissions phases. The sources of uncertainty in the slopes of change in temperature versus cumulative emissions are assessed in terms of their correlations with the varied model parameters. Finally, the extent of continued warming after emissions cease is assessed in terms of the zero emissions commitment and the effect of negative emission applications on reducing any continued warming.

## 2 Theoretical framework

We first introduce the framework under the assumption of only $CO_2$ forcing. A climate metric TCRE (°C $PgC^{-1}$) is defined as the surface warming response to cumulative $CO_2$ emissions

$$TCRE = \frac{\Delta T(t)}{I_{em}}, \tag{1}$$

where $\Delta$ is the change since year 850 CE, $\Delta T(t)$ is the global mean change in surface air temperature (in °C) and $I_{em}(t)$ is the cumulative $CO_2$ emissions (in PgC) from the sum of fossil-fuel emissions and land use changes.

The TCRE may be viewed as a product of two terms, the change in global mean air temperature divided by the change in the atmospheric carbon inventory, $\Delta T(t)/\Delta I_{atmos}(t)$, and the airborne fraction, $\Delta I_{atmos}(t)/I_{em}(t)$, given by the change in the atmospheric carbon inventory (in PgC) divided by the cumulative $CO_2$ emissions (Matthews et al., 2009; Solomon et al., 2009; Gillett et al., 2013; MacDougall, 2016) such that





$$TCRE \equiv \frac{\Delta T(t)}{I_{em}(t)} = \left(\frac{\Delta T(t)}{\Delta I_{atmos}(t)}\right)\left(\frac{\Delta I_{atmos}(t)}{I_{em}(t)}\right), \tag{2}$$

where $\Delta T(t)/\Delta I_{atmos}(t)$ is related to the transient climate response, defined by the temperature change at the time

of doubling of atmospheric $CO_2$ (Matthews et al., 2009). The TCRE is defined in terms of this surface warming response to $CO_2$ forcing, usually following a 1 % annual rise in atmospheric $CO_2$.

Alternatively, the TCRE may be linked to an identity involving a thermal dependence on radiative forcing, defined by the change in temperature divided by the change in radiative forcing, $\Delta F(t)$ (in Wm$^{-2}$), and the radiative forcing dependence on $CO_2$ emissions, defined by the change in radiative forcing divided by the cumulative $CO_2$

emissions (Goodwin et al., 2015; Williams et al., 2016; Williams et al., 2017) such that

$$TCRE \equiv \frac{\Delta T(t)}{I_{em}(t)} = \left(\frac{\Delta T(t)}{\Delta F(t)}\right)\left(\frac{\Delta F(t)}{I_{em}(t)}\right). \qquad . \tag{3}$$

These two viewpoints can be rationalized by rewriting the radiative forcing dependence to $CO_2$ emissions in Eq. 3 in terms of the radiative forcing dependence on atmospheric $CO_2$ and the airborne fraction (Ehlert et al., 2017; Katavouta et al., 2018; Williams et al., 2020).

The TCRE is then defined by the product of the thermal dependence, the radiative dependence between

radiative forcing and atmospheric carbon, and the carbon dependence involving the airborne fraction:

$$TCRE \equiv \frac{\Delta T(t)}{I_{em}(t)} = \underbrace{\left(\frac{\Delta T(t)}{\Delta F(t)}\right)}_{thermal}\underbrace{\left(\frac{\Delta F(t)}{\Delta I_{atmos}(t)}\right)}_{radiative}\underbrace{\left(\frac{\Delta I_{atmos}(t)}{I_{em}(t)}\right)}_{carbon}. \tag{4}$$

The thermal response may be further understood from an empirical global radiative balance (Gregory et al., 2004; Forster et al., 2013). The increase in radiative forcing, $\Delta F(t)$, drives an increase in planetary heat uptake, $N(t)$ (in Wm$^{-2}$), plus a radiative response, which is assumed to be equivalent to the product of the increase in global mean surface air temperature, $\Delta T(t)$, and the climate feedback parameter, $\lambda(t)$ (in °C$^{-1}$ Wm$^{-2}$):

$$\underbrace{\Delta F(t)}_{radiative\ forcing} = \underbrace{N(t)}_{heat\ uptake} + \underbrace{\lambda(t)\Delta T(t)}_{radiative\ response}. \tag{5}$$

The thermal dependence in Eq. 4 given by the dependence of surface warming on radiative forcing, $\Delta T(t)/\Delta F(t)$, is then given by the product of the inverse of the climate feedback, $\lambda^{-1}(t)$, and the planetary heat uptake divided by the radiative forcing, $N(t)/\Delta F(t)$,

$$\frac{\Delta T(t)}{\Delta F(t)} = \frac{1}{\lambda(t)}\left(1 - \frac{N(t)}{\Delta F(t)}\right), \tag{6}$$

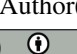



where $1 - N(t)/\Delta F(t)$ represents the fraction of the radiative forcing that warms the surface rather than the ocean interior.

The carbon dependence in Eq. 4 involving the airborne fraction, $\Delta I_{atmos}(t)/I_{em}(t)$, is related to the changes in the ocean-borne, land-borne and sediment-borne fractions (Jones et al., 2013),

$$\frac{\Delta I_{atmos}(t)}{I_{em}(t)} = 1 - \left( \frac{\Delta I_{ocean}(t)}{I_{em}(t)} + \frac{\Delta I_{land}(t)}{I_{em}(t)} + \frac{\Delta I_{sediment}(t)}{I_{em}(t)} \right), \tag{7}$$

where the changes in the ocean, land and sediment inventories are denoted by $\Delta I_{ocean}(t)$, $\Delta I_{land}(t)$ and $\Delta I_{sediment}(t)$ (in PgC), respectively.

The TCRE is formally defined in terms of the climate response to cumulative $CO_2$ emissions following a 1 % annual rise in atmospheric $CO_2$ (Matthews et al., 2009). As the rise in anthropogenic radiative forcing is currently dominated by the radiative forcing from atmospheric $CO_2$, the TCRE is a useful climate metric to understand future climate projections. However, in the more realistic framework we apply here, the warming response includes contributions from non-$CO_2$ forcing. In such experiments, Matthews et al. (2021) advocate estimating the TCRE

by approximately removing the warming due to the non-$CO_2$ radiative forcing by multiplying by a non-dimensional factor $(1 - f_{nc})$, now explicitly acknowledging that $\Delta T(t)$ is not solely driven by $I_{em}(t)$;

$$TCRE = \frac{\Delta T(t)}{I_{em}(t)} (1 - f_{nc}). \tag{8}$$

Matthews et al. (2021) interpret the non-dimensional factor $(1 - f_{nc})$ to represent the non-$CO_2$ fraction of total anthropogenic forcing where $f_{nc} = (\Delta F - \Delta F_{CO})/\Delta F$. This estimation of the TCRE from general forcing scenarios assumes that the time- and scenario- independence of the TCRE translates to a general response

independence from radiative forcing elements.

In order to allow for possible changes in the thermal and carbon responses from the non-$CO_2$ forcing, we prefer to define an effective TCRE (eTCRE) including the effect of the radiative forcing from non-$CO_2$ and $CO_2$ radiative components using a series of mathematical identities (Williams et al., 2016 and 2017), where

$$eTCRE \equiv \frac{\Delta T(t)}{I_{em}(t)} = \underbrace{\left( \frac{\Delta T(t)}{\Delta F(t)} \right)}_{thermal} \underbrace{\left( \frac{\Delta F(t)}{\Delta F_{CO2}(t)} \right) \left( \frac{\Delta F_{CO2}(t)}{\Delta I_{atmos}(t)} \right)}_{radiative\ from\ CO_2\ \&\ non-CO_2} \underbrace{\left( \frac{\Delta I_{atmos}(t)}{I_{em}(t)} \right)}_{carbon}. \tag{9}$$

By including the effect of the non-$CO_2$ radiative forcing, the eTCRE in Eq. 9 is larger than the TCRE with

non-$CO_2$ radiative forcing removed in Eq. 8 whenever the positive radiative effect of non-$CO_2$ greenhouse gases



exceeds the negative effect from aerosols. Our subsequent model diagnostics focus on evaluating the eTCRE and the thermal, radiative and carbon dependences using Eq. 9.

## 3 Methods

### 3.1 GENIE model description and experiment design

The implications of negative emissions technologies deployed for large-scale atmospheric carbon dioxide removal are investigated over 400 years by developing two scenarios of RCP4.5 as the baseline and carbon capture and storage in which annual negative emissions of 2 PgC are applied from year 2070 for 100 years.

The global intermediate complexity Earth system model, GENIE-1 (release 2.7.7) (Holden et al., 2013a) is employed, consisting of the 3-D frictional geostrophic ocean model (GOLDSTEIN) ($36^{\circ} \times 36^{\circ}$ resolution with 16
depths levels in the ocean) coupled to the 2-D energy moisture balance model of the atmosphere (EMBM) and a thermodynamic-dynamic sea-ice model (Edwards and Marsh 2005). The land surface module is the dynamic model of terrestrial carbon and land use change ENTSML (Holden et al 2013a). Ocean biochemistry, deep-sea sediments and rock weathering are modelled by BIOGEM (Ridgewell et al., 2007), SEDGEM ($36^{\circ} \times 36^{\circ}$ resolution) and ROCKGEM (Colbourn et al., 2013) modules, respectively.

The future scenarios were built upon the RCP4.5 GENIE-1 forcing as implemented in Zickfeld et al. (2013). Simulations start from pre-industrial spin ups (Holden et al., 2013b) and follow historical transients forcing from 850 to 2005 CE (Eby et al., 2013). The historical forcing includes $CO_2$ emissions, non-$CO_2$ radiative forcings, and land use changes, including both anthropogenic and natural sources (volcanic eruptions and solar variability). From the year 2005, the model is forced with $CO_2$ emissions consistent with RCP4.5 until 2100 (Meinshausen et
al., 2011), held constant until 2120 and then set to zero for the remainder of the simulation to year 2420 (Fig.1). In the carbon capture and storage (CCS) scenario, $CO_2$ emissions are reduced by 2 PgC from year 2070 to 2170, so applying net negative emissions from 2120 to 2170. In both scenarios, land use change and non-$CO_2$ forcing were held fixed at RCP4.5 values from year 2020. The land use change emissions in Fig. 1 were diagnosed as the difference in land carbon relative to a third 850 to 2420 ensemble that applied RCP4.5 forcing with no land-use
change (i.e. natural vegetation everywhere).





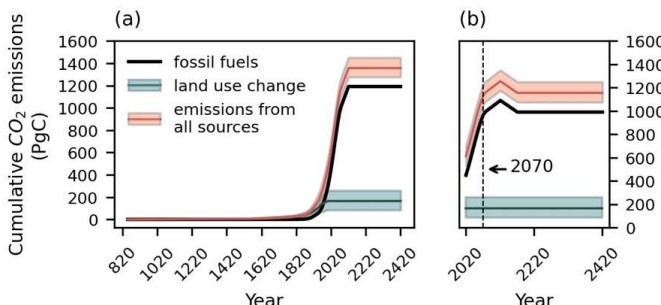

**Figure 1: The cumulative CO₂ emissions from 850 CE till the end of the model integration at year 2420 for (a) RCP4.5 (baseline) and (b) carbon capture and storage scenarios. Solid lines show the median values, and shaded areas indicate the values between the 10th and 90th percentiles. The dashed line shows the beginning of the carbon capture and storage application (year 2070).**

To quantify the uncertainty in climate and carbon-cycle responses, we used an ensemble of 86 members generated from different combinations of 28 model parameters (Foley et al., 2016). These parameters were selected for their importance for climate, ocean dynamics and carbon cycle and create diverse plausible climate states by varying over the entire range of possible inputs rather than the best values (Holden et al., 2013a, 2013b). Eighty-two of the 86 parameter sets successfully completed both simulations, and these 82 members are used in all the subsequent analyses. The ensembles span a wide range of responses at the end of the positive emission phase at year 2120: the increase in surface air temperature ranges from 1.8 to 5 °C; the strength of the Atlantic meridional overturning circulation extends from 6.7 to 24.4 Sv; the land carbon uptake varies from a loss of 94 PgC to a gain of 621 PgC; and the ocean carbon uptake ranges from a gain of 347 to 785 PgC (Fig. 2).

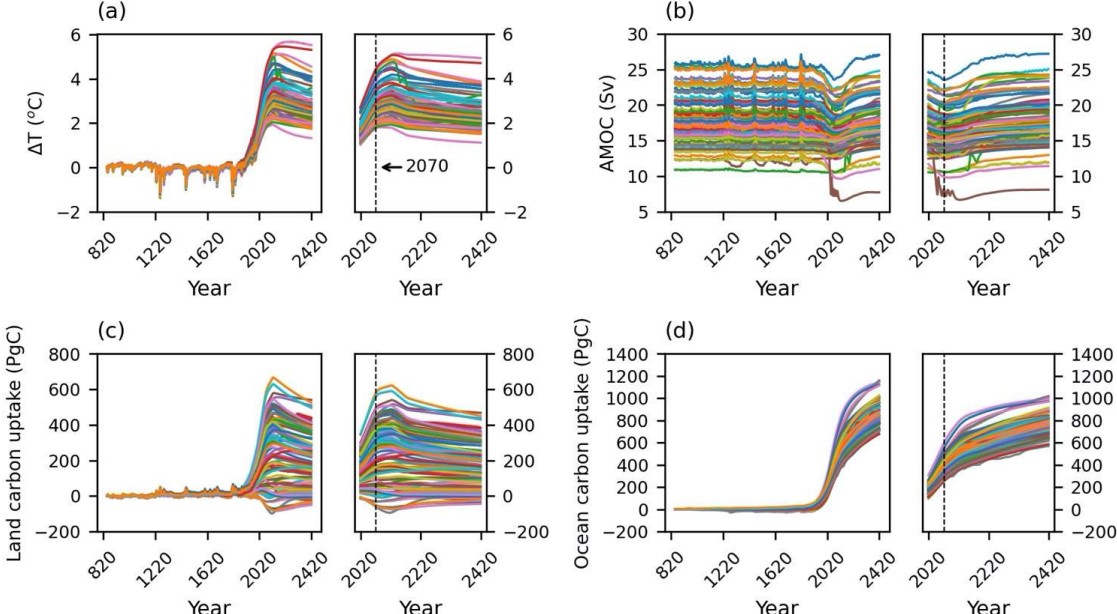

**Figure 2: Inter-model ensemble spread for (a) change in the surface air temperature, (b) Atlantic meridional overturning circulation (AMOC), (c) change in land carbon pool and (d) change in ocean carbon pool from year 850 CE until year 2420 following the RCP4.5 (baseline) (left column), and carbon capture and storage scenarios (right column). The dashed line shows the beginning of the carbon capture and storage application (year 2070).**

### 3.2 Carbon feedback

The distribution of carbon between carbon inventories is diagnosed (Fig. 3), and carbon conservation ensures that at all times the sum of the change in the carbon content of the atmosphere, $\Delta I_{atmos}(t)$, ocean, $\Delta I_{ocean}(t)$, land, $\Delta I_{land}(t)$, and ocean sediment, $\Delta I_{sediment}(t)$, equals the cumulative $CO_2$ emissions from both land use change and fossil fuels, $I_{em}(t)$, with all inventories in PgC,

$$\Delta I_{atmos}(t) + \Delta I_{ocean}(t) + \Delta I_{land}(t) + \Delta I_{sediment}(t) = I_{em}(t) \tag{10}$$

Aside from the ocean sediments, which lose carbon, there is an increase in the carbon content of all inventories between the years 2020 and 2120, the positive emission phase in both the baseline and carbon capture and storage scenarios (Fig. 3). The application of carbon capture and storage from year 2070 decreases the total carbon



inventory until year 2170. During the post-emissions phase, in both scenarios, the increase in ocean storage is associated with a decrease in carbon content in the atmosphere, land and sediment.

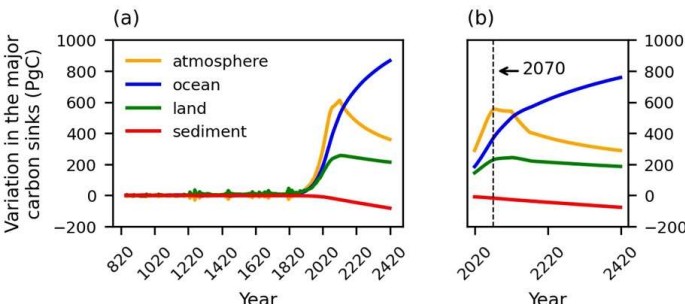

**Figure 3: The ensemble average change in the major carbon inventories from 850 CE until year 2420 (mean of model ensembles) for (a) RCP4.5 (baseline) and (b) carbon capture and storage scenarios. The dashed line shows the beginning of the carbon capture and storage application (the year 2070).**

### 3.3 Thermal feedback

For the thermal analysis, a global energy balance (Eq. 5) is diagnosed, $\Delta F(t) = N(t) + \lambda(t)\Delta T(t)$, in which the
energy balance is expressed as the relationship between radiative forcing, $\Delta F(t)$ (Wm$^{-2}$), planetary heat uptake, $N(t)$ (Wm$^{-2}$) and radiative response, $\lambda(t)\Delta T(t)$ (Wm$^{-2}$).

The radiative forcing, $\Delta F(t)$, is the sum of non-CO$_2$-induced radiative forcing (including land use change albedo) and CO$_2$-induced radiative forcing. For both scenarios, the non-CO$_2$ radiative forcing, $\Delta F_{non-\,2}(t)$ (Wm$^{-2}$) is a prescribed model forcing input, besides land use change which is diagnosed as the change in reflected surface
insolation under land use change relative to that with natural vegetation, averaged annually across all grid cells. The prescribed term, which includes non-CO$_2$ trace gases, volcanic aerosols and anthropogenic aerosols, was fixed to 0.69 Wm$^{-2}$, the value in RCP4.5 at year 2020, for the remainder of the simulations. The land use change maps were also fixed from year 2020 and these were associated with a global forcing of -0.46 to 0.06 Wm$^{-2}$ (25[th] to 75[th] percentile range) and mean and median values of -0.18 and -0.21 Wm$^{-2}$, respectively, across the ensemble. The
uncertainty is driven primarily by crop albedo which varies between 0.12 and 0.18 across the ensemble (Holden et al 2013a). The CO$_2$-induced radiative forcing, $\Delta F_{CO_2}(t)$ (Wm$^{-2}$), was calculated individually for each simulation based on the atmospheric CO$_2$ concentration (C(t) (ppm)) as outlined in (IPCC 2001):

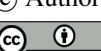



$$\Delta F_{CO_2}(t) = \alpha \, ln\left(\frac{C\,(t)}{C(t_0)}\right) \tag{11}$$

where α is a constant equal to 5.35 Wm$^{-2}$ and C(t$_0$) equals 278 ppm.

The ocean heat uptake is used to estimate the planetary heat uptake, given that 90 % of the Earth's total energy increase is due to the ocean warming (Church et al., 2011). The climate feedback parameter, $\lambda(t)$ (°C$^{-1}$ (Wm$^{-2}$)) is diagnosed from the ocean heat uptake and the change in surface air temperature (Eq. 5). Most of the radiative forcing drives a radiative response involving a rise in surface air temperature, rather than an increase in ocean heat uptake (Fig. 4).

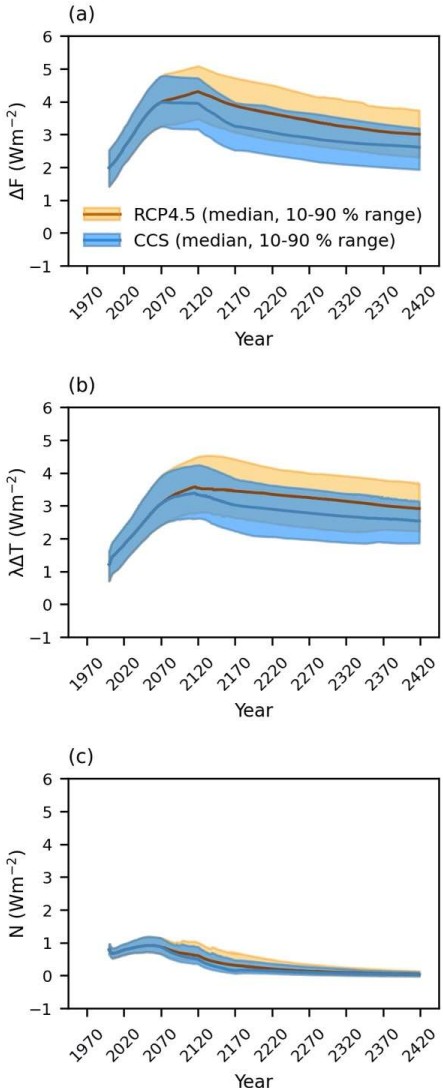

**Figure 4: The evolution of (a) radiative forcing, (b) radiative response and (c) ocean heat uptake in RCP4.5 (baseline), and carbon capture and storage scenarios from year 2000. Solid lines show the median values and shaded areas indicate the values between the 10th and 90th percentiles in baseline (orange) and carbon capture and storage (blue) scenarios.**



**4 The sensitivity of surface air temperature to cumulative CO₂ emissions**

In this section, we evaluate the Earth system response to cumulative $CO_2$ emissions in terms of the thermal,
radiative and carbon feedbacks to understand the reason for the inter-model spread in the slopes of the surface
warming versus cumulative $CO_2$ emissions curve over different emissions phases.

The results of GENIE-1 simulations for both scenarios show a linear relationship between the change in the
surface air temperature and cumulative $CO_2$ emissions over the positive emissions phases (Fig. 5), with the slopes
of this relationship varying between ~1.79 and 3.39 °C EgC⁻¹ (based on the 10 % and 90 % percentile values). The
range of slopes of the $\Delta T$ versus $I_{em}$ curve, calculated by linear regression, over the net negative emissions phase
(i.e. in the carbon capture and storage scenario) is similar to that in the positive emissions phase. During the net
negative emissions phase, the warming relationship is not linear in all ensemble members, and exhibits a hysteresis
behaviour, as previously identified in Zickfeld et al. (2016) and Jeltsch-Thömmes et al. (2020). Differences in the
rates of surface air temperature change over the negative emissions phase are due to the comparable contribution
from the carbon cycle uncertainties in our emissions-forced experiment (discussed in Sect. 4.2.2).

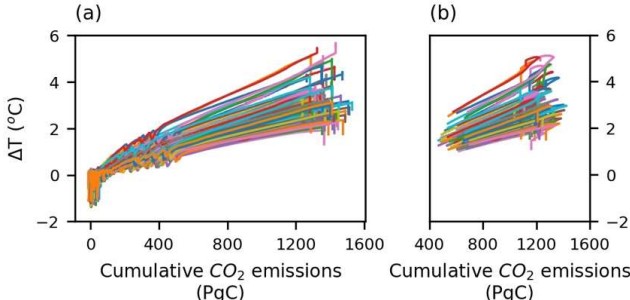

**Figure 5: Change in the surface air temperature versus cumulative CO₂ emissions from 850 CE until year 2420 in (a) RCP4.5 (baseline) and (b) carbon capture and storage scenarios. Note that the panel (b) only shows data from after year 2020.**

**4.1 Drivers of the effective TCRE**

Following Sect. 2, the effective TCRE is evaluated in terms of the product of (i) the dependence of surface warming
on the radiative forcing, referred to as the thermal dependence, $\Delta T(t)/\Delta F(t)$; (ii) the dependence of the radiative
forcing on the cumulative $CO_2$ emissions, referred to as a radiative dependence, $\Delta F(t)/\Delta I_{atmos}(t)$; and (iii) the
airborne fraction, $\Delta I_{atmos}(t)/I_{em}(t)$, referred to as a carbon dependence (Eq. 9):



$$eTCRE \equiv \frac{\Delta T(t)}{I_{em}(t)} = \underbrace{\left(\frac{\Delta T(t)}{\Delta F(t)}\right)}_{thermal} \underbrace{\left(\frac{\Delta F(t)}{\Delta F_{CO2}(t)}\right)\left(\frac{\Delta F_{CO2}(t)}{\Delta I_{atmos}(t)}\right)}_{radiative\ from\ CO_2\ \&\ non-CO_2} \underbrace{\left(\frac{\Delta I_{atmos}(t)}{I_{em}(t)}\right)}_{carbon}.$$

The model ensemble for both baseline and carbon capture scenarios reveals a decrease in eTCRE from the median value of 2.62 °C EgC⁻¹ in year 2020 to 1.96 °C EgC⁻¹ in year 2420 (with 10-90 % range of 1.90 to 3.65 and 1.43 to 2.78 °C EgC⁻¹ respectively) (Fig. 6a). During the positive emission phase (to year 2120) this reduction is

driven by a weakening in the increase in radiative forcing with an increase in atmospheric carbon (Fig. 6b), which dominates over the increase in the thermal dependence (Fig. 6d). During the negative and zero emission phases (from year 2120), the eTCRE reduction is driven by the reducing airborne fraction as $CO_2$ is drawn down by the ocean (Fig. 6e).

The eTCRE is scenario dependent and varies with both $CO_2$ and non-$CO_2$ portions of the total radiative

forcing. Following the analysis of Matthews et al. (2021), we quantify the spread of the non-$CO_2$ fraction of total anthropogenic forcing, $f_{nc}$ (from Eq. 8), between 2020 and 2100 for RCP4.5 as well as the other three RCP scenarios (2.6, 6.0 and 8.5) (Table S1) to investigate the extent of scenario dependency of the eTCRE. The results showed that the change in $f_{nc}$ across all RCP scenarios and times ranges from ~6 % to ~17 % (25 to 75 % range) with the mean and median value of ~11 % (Table S1). The results could be in part due to the fixed non-$CO_2$

radiative forcing from 2020 onwards in the $f_{nc}$ calculations which diminishes the effect of non-$CO_2$ radiative forcing. The TCRE diagnosed by removing the non-$CO_2$ warming factor (from Eq. 8) varies from 1.6 to 2.8 °C EgC⁻¹ (10 to 90 % range) with a median value of 2.2 °C EgC⁻¹ between the years 2020 to 2100 (Fig. S1).

The uncertainty in the eTCRE, and its dependencies for the model ensemble, is assessed based on the nondimensional coefficient of variation, given by the inter-model standard deviation divided by the ensemble mean

(Williams et al., 2020). The uncertainty in the eTCRE varies from 0.23 to 0.27 over the course of the model integrations and is marginally larger by 0.01 for the negative emissions (Table S2).

In both scenarios, the coefficients of variation for the thermal dependence and airborne fraction provide the dominant contributions to the eTCRE uncertainty, with their values ranging from 0.17 to 0.20 and 0.18 and 0.21 respectively (Table S2). These contributions are larger than the coefficient of variation for the fractional radiative

forcing contribution from atmospheric $CO_2$, $\Delta F(t)/\Delta F_{CO2}(t)$, ranging from 0.11 to 0.14, and the dependence of the radiative forcing on atmospheric $CO_2$, $\Delta F_{CO2}(t)/\Delta I_{atmos}(t)$, only ranging from 0.04 to 0.05.

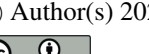

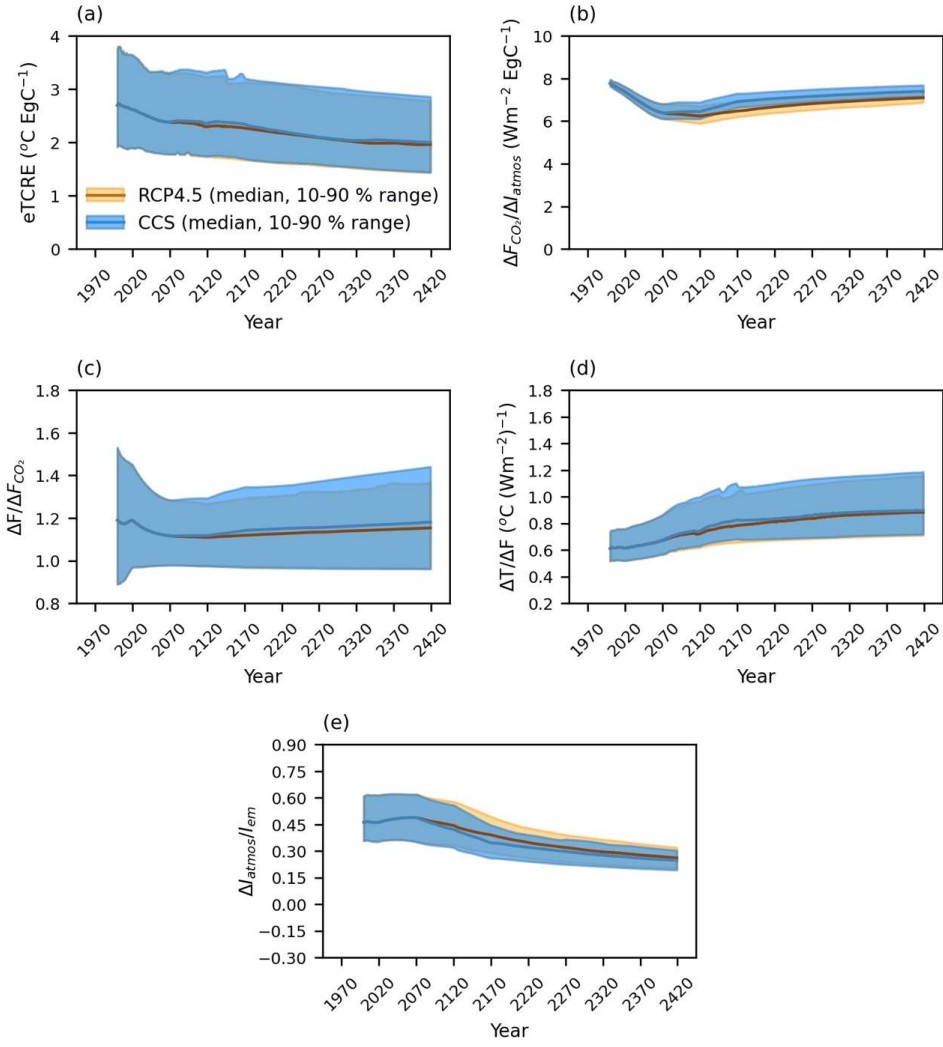

**Figure 6: Effective transient climate response to the cumulative CO₂ emissions and its components for RCP4.5 (baseline) and carbon capture and storage (CCS) scenarios from year 2000. (a) Effective TCRE (eTCRE), (b) fractional radiative forcing contribution from atmospheric CO₂, (c) dependence of the radiative forcing on atmospheric CO₂, (d) airborne fraction and (e) thermal dependence. Solid lines show the median values, and shaded areas indicate the values between the 10th and 90th percentiles in baseline (orange) and carbon capture and storage (blue) scenarios.**






### 4.1.1 Carbon dependence for the effective TCRE

The fraction of emitted $CO_2$ that remains in each carbon inventory (based on Eq. 7) varies over the course of the

integration. The carbon dependence for the eTCRE is given by the airborne fraction of carbon emissions, $\Delta I_{atmos}(t)/I_{em}(t)$. In both scenarios, the airborne fraction strengthens by ~7 % (based on the median values) from years 2020 to 2070 (Fig. 7a), likely as a result of increasing $CO_2$ emissions and weakening terrestrial carbon sinks. After year 2070 and a cessation of $CO_2$ emissions, the ocean becomes the dominant carbon sink with an increase in the ocean-borne fraction, $\Delta I_{ocean}(t)/I_{em}(t)$, to ~65 % (median value) by 2420 (Fig. 7b). The land-

borne fraction, $\Delta I_{land}(t)/I_{em}(t)$ decreases from ~0.24 (median value) in 2020 to the minimum value of ~16 % in 2420 (Fig. 7c). The sediment-borne fraction, $\Delta I_{sediment}(t)/I_{em}(t)$, remains negative at ~-0.03 (median value) over the entire period (Fig. 7d), and therefore acts as a weak carbon source.

The coefficient of variation is the largest for the land-borne fraction (~0.7), followed by the sediment-borne fraction (~-0.5) and then the airborne and ocean-borne fractions (~0.2) (Table S3), implying that the land carbon

system provides the main contribution to the model ensemble spread.



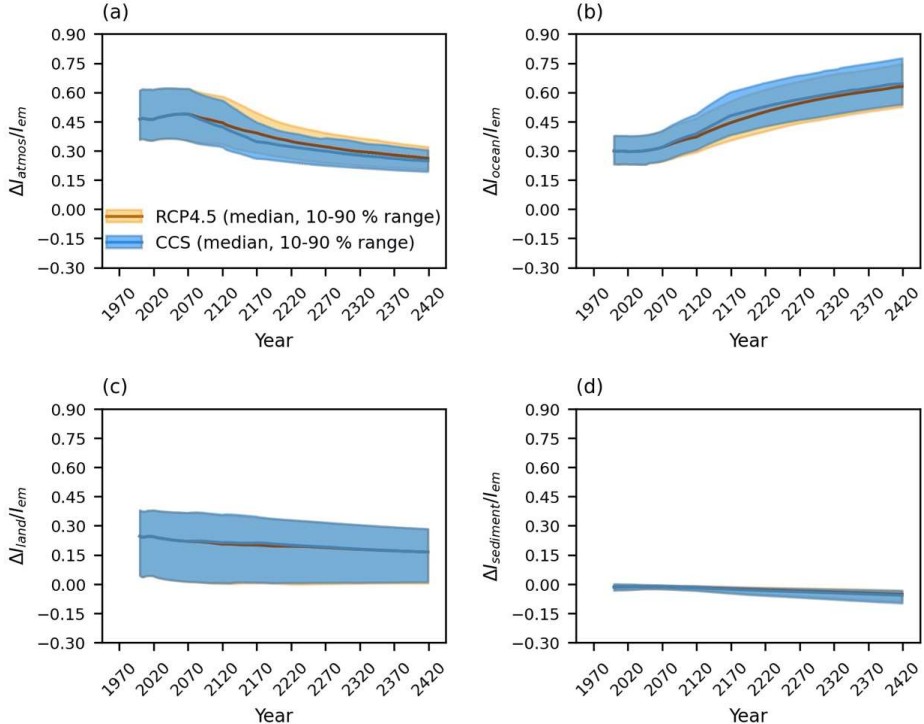

**Figure 7: The evolution of (a) airborne fraction, (b) ocean-borne fraction, (c) land-borne fraction and (d) sediment-borne fraction in RCP4.5 (baseline) and carbon capture and storage (CCS) scenarios. Solid lines show the median values, and shaded areas indicate the values between the 10th and 90th percentiles in baseline (orange) and carbon capture and storage (blue) scenarios from year 2000.**


### 4.1.2 Radiative forcing dependence on atmospheric CO$_2$ for the effective TCRE

By year 2120, the radiative forcing dependence on atmospheric CO$_2$ emissions, $\Delta F(t)/\Delta I_{atmos}(t)$, weakens due to a saturation in the radiative forcing with an increase in atmospheric CO$_2$ (Gillett et al. 2013; William et al., 2020) (Fig. 8a-b). Over the next few centuries from year 2120 onwards, $\Delta F(t)/\Delta I_{atmos}(t)$ rises again due to a

decrease in atmospheric CO$_2$ associated with the decrease in the airborne fraction (Fig. 8c).





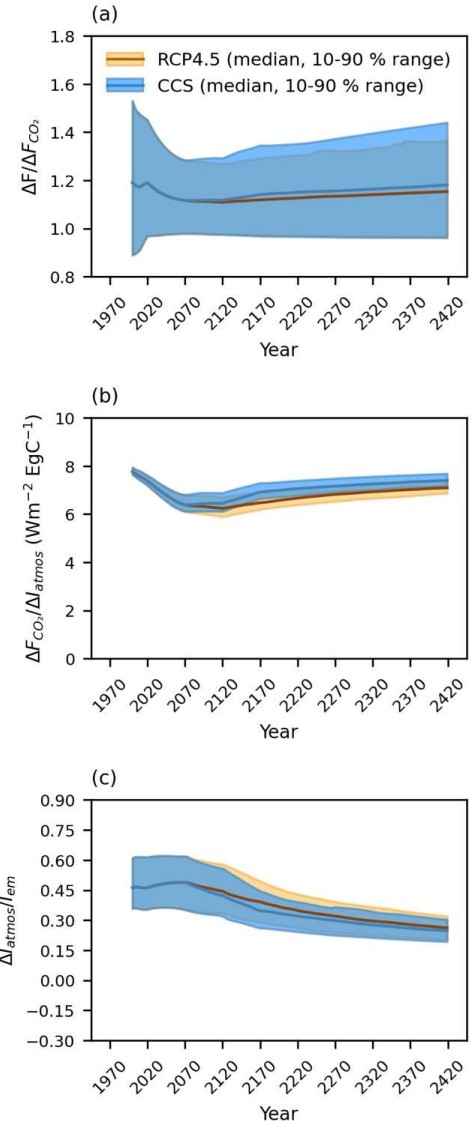

**Figure 8: Radiative forcing dependence for the effective TCRE and its components in RCP4.5 (baseline) and carbon capture and storage (CCS) scenarios from year 2000. (a) Fractional radiative forcing contribution from atmospheric CO₂, (b) dependence of the radiative forcing on atmospheric CO₂ and (c) airborne fraction. Solid lines show the median values, and shaded areas indicate the values between the 10th and 90th percentiles in baseline (orange) and carbon capture and storage (blue) scenarios.**






### 4.1.3 Thermal dependence for the effective TCRE

For both scenarios, the thermal dependence of the eTCRE, involving the dependence of the surface warming on the radiative forcing, $\Delta T(t)/\Delta F(t)$, increases in all emissions phases (Fig. 9a) due to the reinforcing contributions
of the inverse of the climate feedback parameter, $\lambda(t)^{-1}$ (Fig. 9b) and the fraction of the radiative forcing warming the surface, $1 - N(t)/\Delta F(t)$ (Fig. 9c). The increase in $\lambda(t)^{-1}$ is equivalent to a slight decrease in the climate feedback $\lambda(t)$. The temporal evolution of the climate feedback parameter is mirrored in other climate model studies as climate feedbacks evolve on different timescales for a myriad of reasons (e.g., Gregory et al. 2004; Armour et al., 2013; Knutti and Rugenstein, 2015; Goodwin, 2018). The fraction of the radiative forcing warming
the surface increases by ~30 % (based on the median values) from years 2020 to 2420 and with a corresponding reduction in the heat transfer into the deep ocean; by year 2420, nearly all the radiative forcing is warming the surface with the ratio $1 - N(t)/\Delta F(t)$ reaching 0.99 (median values) (Fig. 9c-d). This response is probably due to an increase in ocean stratification from the rise in surface ocean temperature (Figs. S2-S4) from the increased radiative forcing.

In both scenarios with and without carbon capture and storage, the coefficient of variation for the thermal dependence remains around 0.2 over the future scenarios. Within the thermal dependence, the term relating to the climate feedback parameter $\lambda(t)^{-1}$ has a coefficient of variation more than twice that of the fraction of the radiative forcing warming the surface $1 - N(t)/\Delta F(t)$ (0.24 versus 0.1, Table S2). As the thermal dependence terms $\lambda(t)^{-1}$ and $1 - N(t)/\Delta F(t)$ are strongly anti-correlated (Fig. S5), the relative spread in the thermal
response is thus mitigated by the feedback between the climate feedback parameter and the fraction of the radiative forcing warming the surface.



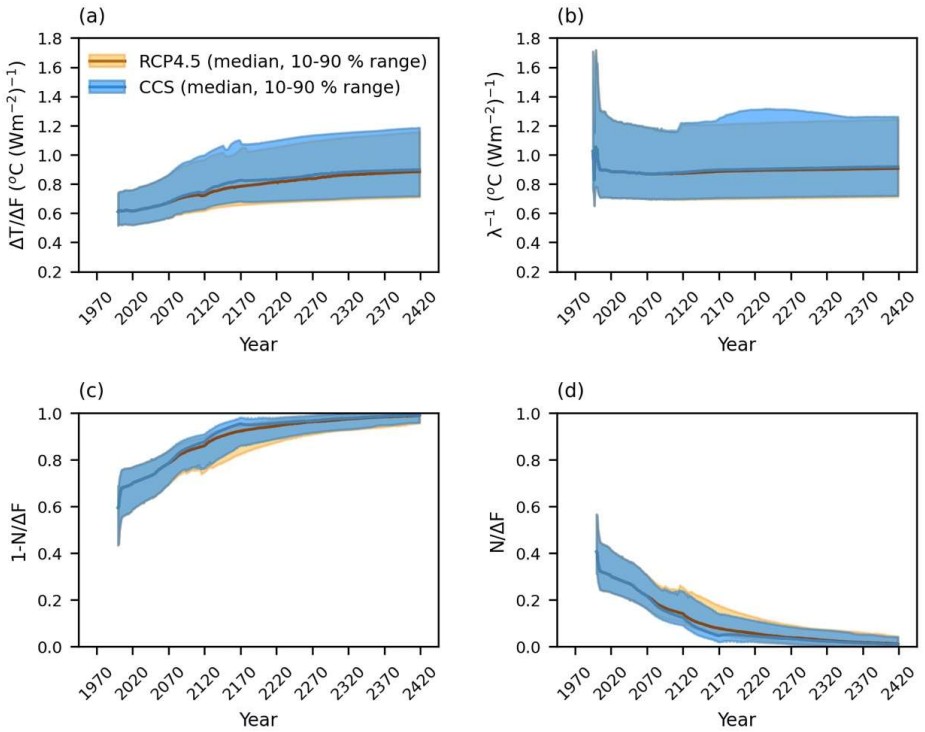

**Figure 9: The evolution of (a) thermal dependence for the effective TCRE given by the dependence of the surface warming on the radiative forcing, $\Delta T(t)/\Delta F(t)$, and the contributions from (b) the inverse of the climate feedback, (c) the fraction of the radiative forcing warming the surface and (d) the fraction of the radiative forcing warming the ocean interior in RCP4.5 (baseline) and carbon capture and storage scenarios from year 2000. Solid lines show the median values, and shaded areas indicate the values between the 10th and 90th percentiles in baseline (orange) and carbon capture and storage (blue) scenarios.**

### 4.2 The asymmetry of the Earth system response to positive and negative emissions

### 4.2.1 Hysteresis

The relationship between the surface air temperature and atmospheric $CO_2$ exhibits hysteresis behaviour in most ensemble members, consistent with climate change reversibility studies (Fig. 10a-b) (Tokarska and Zickfeld, 2015; Jeltsch-Thömmes et al., 2020). The temperature remains at high levels after high atmospheric $CO_2$ concurrent with a decrease in the ocean heat uptake, *N(t)* (Fig. 10c-d). The ability of the ocean interior in taking up heat diminishes in time, probably due to increasing stratification and weakening ventilation. The fraction of the radiative forcing





warming the ocean interior, $N/\Delta F(t)$ (Fig. 10e-f) then continues to decrease after the peak in atmospheric $CO_2$ leading to higher surface air temperatures even after the lower $CO_2$ concentrations are restored.

In the carbon capture and storage scenario, the atmospheric $CO_2$ declines during the negative emissions phase from year 2070 (Fig. S6) (associated with the cumulative $CO_2$ emissions of ~1050 PgC (Figs. 1 and 11b). After the cessation of the emissions, the atmospheric $CO_2$ continues to decrease in both scenarios (Fig. 11a-b) mainly

due to uptake by the ocean and to a lesser extent the land (Fig. 11c-f). The ocean carbon uptake is governed by the air-sea flux of $CO_2$ and thermocline ventilation, with uncertainties dominated by ventilation processes transferring carbon from the surface ocean to the main thermocline and deep ocean (Holden et al 2013b, Goodwin et al., 2015; Zickfeld et al., 2016; Jeltsch-Thömmes et al., 2020). The ocean continues to take up carbon after the peak in atmospheric $CO_2$ as there is continuing long-term adjustment and ventilation of the deep ocean (Fig. 11c-d). The

complex responses of land carbon (Fig. 11e-f) are driven by a range of competing processes, most notably carbon uptake through $CO_2$ fertilization and the carbon release through historical land use changes and accelerated respiration under warming.






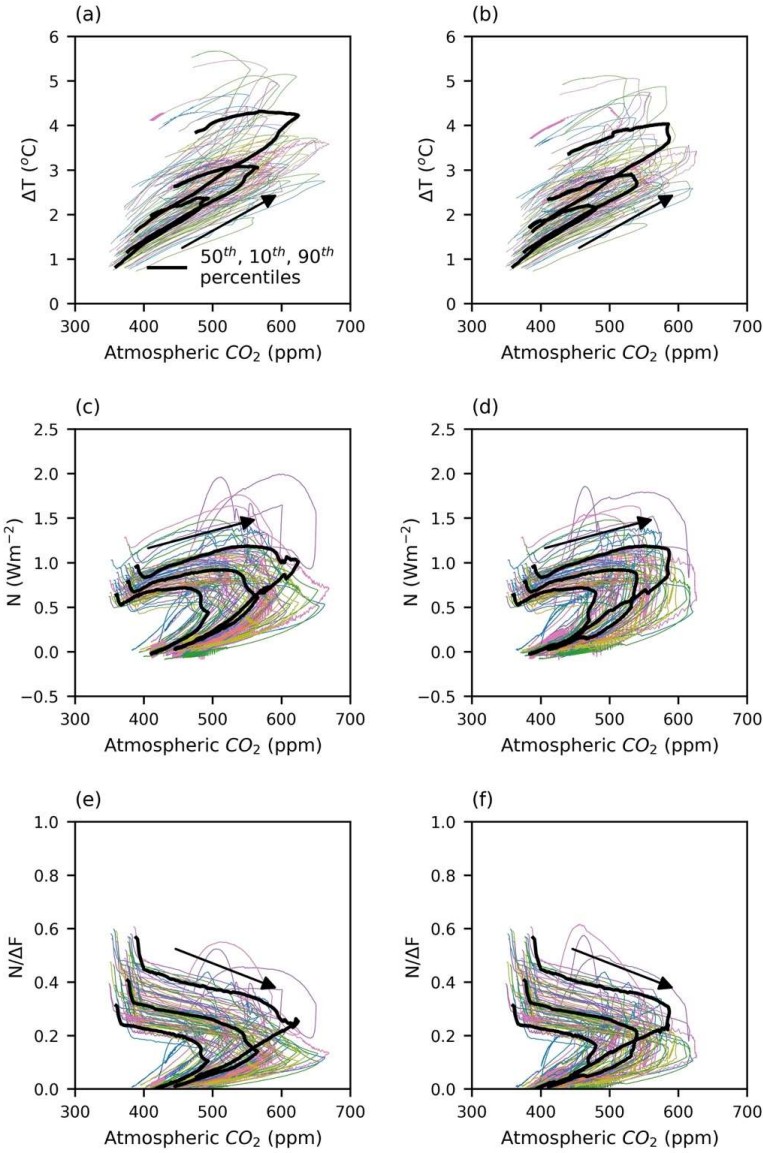

**Figure 10: The thermal variables versus atmospheric $CO_2$ in RCP4.5 (baseline) (left column) and the carbon capture and storage scenarios (right column) from year 2000: (a), (b) Change in surface air temperature; (c), (d) ocean heat uptake; and (e), (f) fraction of the radiative forcing warming the ocean interior. In each panel the solid black lines show the median, 10th and 90th percentile values of the atmospheric $CO_2$ on the x-axis versus the median, 10th and 90th percentile values of the thermal variables on the y-axis.**



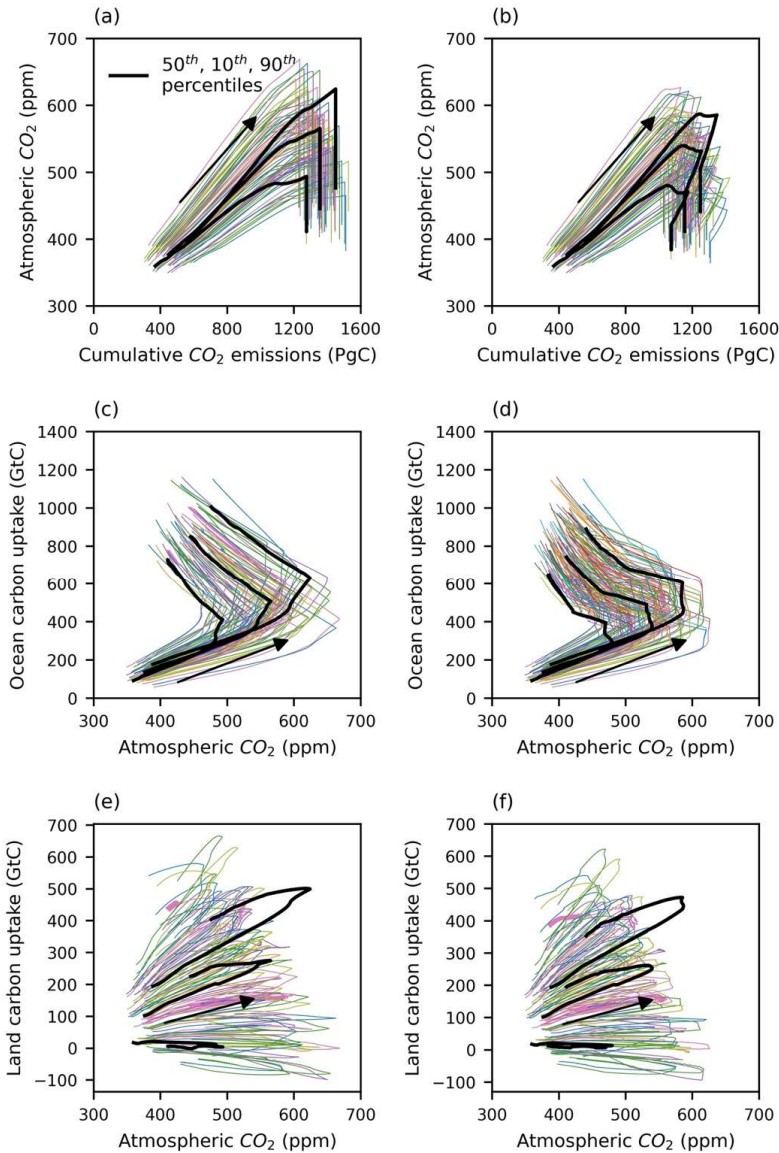

**Figure 11: The carbon variables versus atmospheric $CO_2$ in RCP4.5 (baseline) (left column) and the carbon capture and storage scenarios (right column) from year 2000: (a), (b) Cumulative $CO_2$ emissions;(c), (d) change in the ocean carbon pool; and (e), (f) change in the land carbon pool. In each panel the solid black lines show the median, 10th and 90th percentile values of the atmospheric $CO_2$ on the x-axis versus the median, 10th and 90th percentile values of the carbon variables on the y-axis.**



### 4.2.2 Correlation between the model parameters and the slope of the change in surface air temperature versus cumulative CO₂ emissions ($\Delta T/\Delta I_{em}$)

We calculated the coefficients of determination ($R^2$) between $\Delta T/\Delta I_{em}$ and the 28 model parameters across the ensemble during both positive and net negative emissions phases. For this purpose, four of the 82 simulations were omitted as outliers because they were undergoing substantial re-organisation of ocean circulation during the period of negative emissions (Fig. S7), significantly perturbing ocean heat uptake.

During the positive emissions phase, uncertainty in $\Delta T/\Delta I_{em}$ is dominated by the radiative feedback
parameter (OL1) ($R^2\sim$64 %) (Table 1), which perturbs outgoing longwave radiation proportionally to $\Delta T$ (Matthews and Caldeira, 2007). This parameter is primarily designed to capture unmodelled cloud responses to global average temperature change, and it has previously been shown to drive 81 % of the variance in GENIE-1 climate sensitivity (Holden et al., 2010). Although radiative forcing uncertainty dominates, carbon cycle parameters also drive $\Delta T/\Delta I_{em}$ variance via the land use change soil carbon parameter (KC) ($R^2\sim$12 %) through
its control on soil carbon losses under land use change that continue after land use change is held fixed from 2020 due to the long (multi-decadal) soil time scales. The fractional vegetation parameter (VFC) ($R^2\sim$10 %) drives additional carbon cycle uncertainty through its control on terrestrial carbon surface density.

During net negative emissions within the carbon capture and storage experiment (2120-2170), uncertainty in $\Delta T/\Delta I_{em}$ is affected again by the radiative feedback parameter (OL1, ~15 %) but now also by the effects of ocean
transport and the carbon cycle. In the ocean, uncertainty is dominated by the wind stress scaling parameter (WSF, ~16 %), which drives circulation strength and is the dominant control of uncertainty in ocean carbon uptake (Holden et al 2013b). On land, the dominant control is via CO₂ fertilisation (VPC) (~14 %), a major source of terrestrial carbon uncertainty.






**Table 1: Coefficient of determination $R^2$ between model parameters and $\Delta T/\Delta I_{em}$ in RCP4.5 (baseline) and carbon capture and storage scenarios over different emissions phases based on the coefficients of determination ($R^2$) (%). $R^2 > 50$ % denotes strong correlation, and $R^2 > 10$ % moderate correlation. The values less than 10 % are shown in Table S4.**


| Emissions phase | Parameter | Description | baseline | CCS |
|---|---|---|---|---|
| 2020-2120 | OL1 | Radiative feedback parameter | 63.7 | 63.0 |
| | KC | Land use change soil carbon | 12.0 | 12.3 |
| | VFC | Fractional vegetation dependence on vegetation carbon density | 9.9 | 10.1 |
| 2120-2170 | WSF | Wind-scale factor | - | 16.4 |
| | OL1 | OLR feedback parameter | - | 14.6 |
| | VPC | $CO_2$ fertilisation | - | 14.0 |

### 4.3 The Zero Emissions Commitment

The climate response after net zero emissions is an important climate metric, encapsulated in the zero emissions commitment (ZEC) given by the mean surface air temperature change after $CO_2$ emissions cease (Hare and

Meinshausen, 2006; Matthews and Caldeira, 2008, Froelicher and Paynter, 2015; MacDougall et al., 2020). Quantification of the ZEC is critical for calculating the remaining carbon budget. Whether there is continued surface warming depends on a competition between a cooling effect from the reduction of the radiative forcing from atmospheric $CO_2$ as carbon is taken up by the ocean and terrestrial biosphere versus a surface warming effect from a decline in the heat uptake by the ocean interior (Williams et al., 2017b). In an analysis of Earth system

model responses, MacDougall et al (2020) found that the model mean for the ZEC was close to zero, but that there was a wide spread in continued warming and cooling responses from individual models.

Our baseline experiment, which applies RCP4.5 $CO_2$ emissions until year 2120 and zero emissions thereafter, with cumulative emissions of ~1360 GtC (median value), is approximately comparable to the 1000 PgC experiment of the multi-model intercomparison of MacDougal et al (2020), in which emissions were derived to

drive a 1 % $yr^{-1}$ rise in atmospheric $CO_2$ concentration from pre-industrial until cumulative emissions of 1000 GtC.

In the baseline experiment, when emissions cease the surface temperature continues to rise in 50 % of the ensemble members (the upward vertical lines in Fig. 5). Following the analysis of MacDougall et al. (2020), we define $ZEC_{25}$, $ZEC_{50}$, $ZEC_{90}$ as the mean surface air temperature anomalies (relative to the year that emissions cease) at the 25th, 50th and 90th years after the cessation of emissions, to account for the implications of ZEC over



a range of multi-decadal times scales relevant to climate policy. Diagnosed ZEC values are illustrated in Fig. 12 (baseline plotted as orange bars). In the baseline, the distribution of the $ZEC_{25}$ and $ZEC_{50}$ display an uncertain sign. There is a temperature overshoot in 50 % of $ZEC_{25}$ values (10-90 % range from -0.07 to 0.05 °C) and in 24 % of $ZEC_{50}$ values (range from -0.14 to 0.04 °C). The values of ZEC decrease over time so that the $ZEC_{90}$ remains at or below zero in all ensemble members, ranging from -0.26 to 0.00 °C.  The ensemble means of $ZEC_{25}$, $ZEC_{50}$

and $ZEC_{90}$ are -0.01, -0.06 and -0.15 °C respectively, consistent with values of -0.01, -0.07 and -0.12 °C in the 1000 PgC experiment of MacDougall et al (2020). We note that our uncertainties are lower than MacDougall et al. (2020), which at least in part reflects the absence of internal (decadal) variability in the EMBM of GENIE-1, noting that inter-annual, but not decadal, variability was removed from MacDougall et al. (2020) through 20-year averaging.

We additionally consider ZEC metrics for the ensemble including carbon capture and storage. In contrast to the baseline, surface temperatures decrease in all the ensemble members after cessation of positive emissions. We consider two alternative interpretations of the ZEC, the warming after the cessation of net positive emissions (in 2120) and the warming after the cessation of net negative emissions (in 2170). The former may be more relevant from a policy perspective (as the time of likely peak warming), while the latter is theoretically useful to quantify

committed warming when emissions are precisely zero.

The blue bars in Fig. 12 illustrate the ZEC results for the carbon capture and storage scenario from 2170. Ensemble means are -0.08 °C, -0.13 °C and -0.19 °C for $ZEC_{25}$, $ZEC_{50}$ and $ZEC_{90}$, respectively. Notably, the values are robustly negative, ranging from -0.12 to -0.04 °C for $ZEC_{25}$, -0.19 to -0.07 °C for $ZEC_{50}$, and -0.31 to -0.09 °C for $ZEC_{90}$ (10th-90th percentile range). The green bars in Fig. 12 illustrate the ZEC values from 2120 (which

includes the period of ongoing net negative emissions). The average values of ZEC are significantly lower than from 2170, being -0.11 °C, -0.26 °C, and -0.37 °C, due to the additional cooling driven by net negative emissions. All ZEC values are again robustly negative, varying between -0.14 and -0.05 for $ZEC_{25}$, -0.33 and -0.14 for $ZEC_{50}$, and -0.49 and -0.2 for $ZEC_{90}$ (10 %-90 % percentile values), confirming that no ensemble member exhibits a temperature overshoot after the cessation of positive emissions in the carbon capture and storage scenario.






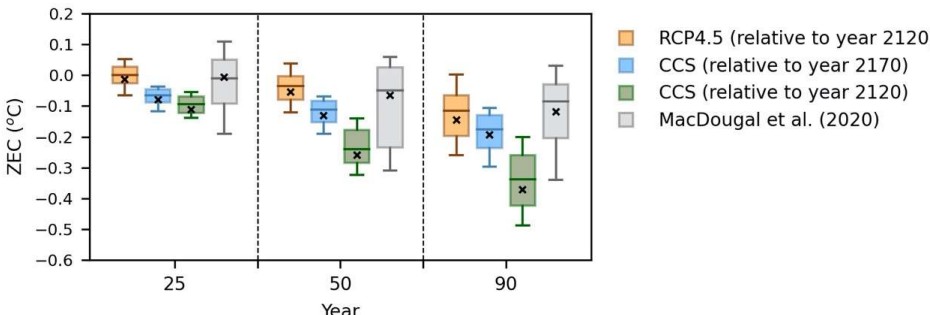

**Figure 12: The distribution of the zero emissions commitment (ZEC) in RCP4.5 (baseline) at 25th, 50th and 90th years relative to year 2120 (orange bars) and in the carbon capture and storage scenario relative to year 2170 (blue bars) and relative to year 2120 (green bars) versus the results of MacDougal et al. (2020) (grey bars). The mean values are shown with cross marks. Note that the year 2120 is the end of the (net) positive emissions phase in both scenarios, and the year 2170 is the end of the net negative emissions phase in the carbon capture and storage scenario.**


## 5 Conclusions

To remain within the Paris climate agreement, there is an increasing need to develop and implement carbon capture and sequestration techniques. However, it is unclear how these effective negative emissions affect the climate

response, as represented by two key climate metrics: the effective TCRE, defining the relationship between surface warming and cumulative $CO_2$ emissions, and the ZEC, defining the anticipated warming after $CO_2$ emissions cease. The effect of negative emissions is assessed here using a large GENIE-1 ensemble, following RCP4.5 as the baseline case, and then including carbon capture and storage as an alternative scenario (with the $CO_2$ removal rate of 2 PgCyr$^{-1}$ over 100 years). The model responses include 82 members that span a wide range of climate and

carbon cycle feedback strengths. This large ensemble analysis is enabled by employing low resolution and intermediate complexity, with most notable simplifications of the fixed wind-field energy-moisture balance atmosphere, neglecting dynamic atmosphere-ocean feedbacks, and the simple model of terrestrial carbon, which neglects nutrient limitation, does not represent permafrost (or methane), and has a 1-level description of soil carbon.

The effective TCRE decreases in time in scenarios with and without carbon capture due to a combination of the weakening in the radiative forcing with an increase in atmospheric carbon during positive emissions and with a reduction in the airborne fraction after emissions cease, which together outweigh the strengthening thermal dependence. The controls on the effective TCRE are similar in model integrations both with and without carbon



capture and storage. This similar response implies that information on the behaviour of early 21[st] century warming
can be extended to projections with moderate amounts of carbon capture and storage.

The comparison of the coefficient of variation for the effective TCRE and its dependencies show that the thermal dependence and airborne fraction almost equally contribute to the uncertainty in the effective TCRE. Our results differ from the analysis of CMIP6 ensembles in which the radiative forcing response and thermal response were the main contributors to the uncertainty in the TCRE (Williams et al., 2020). These inferences are consistent
with a model parameter correlation analysis attributing the weakening in warming slopes versus emissions to radiative feedbacks during net positive emissions, and also affected by changes in the airborne fraction of $CO_2$ during the negative and zero emission phases

The relationship between thermal and carbon feedbacks with an increase in atmospheric $CO_2$ exhibits hysteresis behaviour. The fraction of the radiative forcing warming the surface continues to increase after peak
atmospheric $CO_2$ as the ocean is stratified, leading to higher surface air temperatures after lower atmospheric $CO_2$ values are restored. The increase in the ocean and terrestrial carbon storage after the peak in atmospheric $CO_2$ is associated with the long-term response of each of these carbon sinks as well as the carbon storage from their past carbon uptake.

The zero emission commitment given by the warming after net emissions is close to zero, in the model mean
of integrations excluding carbon capture, the ZEC is -0.01 ºC at 25 years and decreases in time to -0.15 °C at 90 years after emissions cease. However, in this case without carbon capture, the distribution of ZEC after 25 years from the cessation of emissions shows warming in ~50 % of ensemble members. Including modest levels of carbon capture and storage avoids this continued warming after net zero with all ensemble members exhibiting a ZEC close to or below zero. Hence, implementing net negative emissions is required to reduce the risk of over-shoot
and continued warming after net zero is reached, and so increase the probability of meeting the Paris targets. Negative emissions technologies with naturally long $CO_2$ removal lifetimes, such as agricultural enhanced rock weathering (Beerling et al., 2020) may be especially well suited for this purpose as the legacy effects of the repeated application of this technology increase the rate of carbon drawdown per unit area for years after implementation at no incremental cost (Beerling et al., 2020; Vakilifard et al., 2021).




*Data availability.* The data that support the findings of this study are available from the corresponding author upon reasonable request.

*Author contributions.* NV undertook model experimental design, all simulations, and analyses. All authors were
involved in the design of the model experiments, led by NV and RGW. All authors contributed to writing, led by NV and RGW.

*Competing interests.* The authors declare that they have no conflict of interest.

*Acknowledgments.* The authors acknowledge funding with a Leverhulme Research Centre Award, RC-2015-029, from the Leverhulme Trust. RGW is supported by a UK Natural Environment Research Council grant NE/T007788/1.

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
