# Peer review of "Assessment of negative and positive CO2 emissions on global warming metrics using an ensemble of Earth system model simulations"

_Biogeosciences, 2022_

## Author Comment (AC2)

Dear Dr. Bond-Lamberty,

Thank you for the opportunity to resubmit a revised copy of our manuscript, "Assessment of negative and positive $CO_2$ emissions on global warming metrics using a large ensemble of Earth system model simulations". We would also like to thank the reviewers for their valuable time and constructive feedback.

Following this letter are our point-by-point responses to each of the comments of the reviewers. We have uploaded a copy of the original manuscript marked with all the changes made during the revision process. It is our belief that the manuscript has been substantially improved after making the suggested edits. We therefore hope that you will find our revisions in the manuscript and accompanying responses acceptable for publication in the journal of *Biogeosciences*.

Sincerely,
Negar Vakilifard, Corresponding author

*Detailed response to the reviewers' comments*

*Reviewer #1*

1.1.  General Comments

This contribution assesses the benefits of negative emissions / $CO_2$ removal technologies deployment for future climate states using an ensemble of intermediate complexity earth system model results. The authors use effective transient climate response to cumulative $CO_2$ emissions (eTCRE) and zero emissions commitment (ZEC) as metrics to quantify these impacts. The authors find that thermal dependence and airborne fraction of $CO_2$ contribute almost equally to the uncertainty in eTCRE, which is in contrast with recent analysis of the CMIP6 ensemble. Additionally, the authors find that negative emissions deployment can help avoid continued warming after net-zero emissions are reached. The manuscript is clear and well-written and the analysis appears free of errors. However, I have several recommendations aimed at increasing the impact and clarity of this work, which are detailed below.

**We thank the reviewer for their supportive statements on the quality of the paper. We have taken the reviewer's recommendations on board through the revisions made in response to comments 1.2 and 1.3 to increase the impact and clarity of the current work.**

1.2.  Specific Comments

1.2.1. The authors use the RCP 4.5 medium-level mitigation scenario as a benchmark to assess the future climate response to negative $CO_2$ emissions. However much of the discussion of prospective large-scale negative emissions deployment in the recent literature focuses on their use towards limiting end-of-century warming to well-below 2 C, more consistent with RCP 2.6 or RCP 1.9. Although even the "medium" mitigation scenario may seem optimistic relative to the present real-world trajectory, using one or both of these forcing scenarios representing even deeper levels of mitigation could increase the impact of this work. I recommend the authors run similar analysis on one or both of these deeper mitigation scenarios, even as a sensitivity case. This could allow the modeling community, policymakers, and other stakeholders insight into what a "best case" scenario might look like in terms of transient climate response and committed warming.

**Thank you. In order to address this and also comment 2.2.2, we replaced RCP4.5 and CCS scenarios with shared socioeconomic pathways (SSP)1-2.6 as described in Riahi et al. (2017) and Meinshausen et al. (2020), and revised the manuscript based on the results of the simulations for this scenario. We decreased the number of scenarios to one as SSP1-2.6 has been designed based on the realistic future socio-economic developments (O'Neill et al. 2017) and already considers the application of the negative emissions technology, which gives three phases of positive, negative and zero emissions. We explained the model set-up for this scenario in detail in section 3.1.**

1.2.2. In the final paragraph of the conclusions section the authors refer to the need for negative emissions technologies that have naturally long storage times. In the main body of the manuscript it would be helpful to describe exactly what types of carbon removal technologies are represented in the models used to develop the ensemble. Long-lived and permanent storage such as direct air capture or enhanced rock weathering referred to in the conclusions? Or biospheric such as afforestation? Or is the representation of negative emissions agnostic as to the source in the models? Any biospheric contribution to the negative emissions and potential feedbacks or the limitations in representing them should be identified in the discussion around land carbon.

**Thank you for bringing this up. We implement the negative emissions as the reduction in anthropogenic CO$_2$ emissions and assume the carbon leaves the system permanently. This can be the representation of negative emissions technologies such as carbon capture and storage or enhanced rock weathering (ERW), which was shown to permanently remove the atmospheric CO$_2$ (Vakilifard et al., 2021). From a modelling perspective, however, the set-up for the two scenarios slightly differs. For the ERW, it is required to explicitly incorporate the effects of its by-products such as bicarbonate on ocean biogeochemistry to account for its co-benefits for the ocean and marine ecosystems. We referred to these points in section 3.1 of the revised manuscript.**

1.3.    Technical Corrections

The authors should be more precise in differentiating point source carbon capture from carbon capture from the atmosphere for negative emissions. Throughout the manuscript the more generic "carbon capture" or "carbon capture and storage" is used. While readers might infer from context that this is referring to negative emissions, this term should be clearly defined at every use to avoid the possibility of misinterpretation.

**Thank you. In the revised manuscript, we mainly referred to the scenario as SSP1-2.6 rather than carbon capture and storage; however, where applicable we changed the carbon capture to carbon capture and storage to avoid the possibility of misinterpretation for the readers. Please see the abstract and sections 3.1 and 5.**

2.1.    Overall Assessment

The study uses an intermediate complexity Earth system model to assess eTCRE, ZEC and the effect of net negative emissions on these metrics. In addition, the study uses the theoretical framework of Williams et al., 2016 to decompose TCRE into components. Although the perturbed parameter model set-up is a promising method, the experiment design is baffling and thus the wider meaning of the results is very unclear. Additionally, the paper focuses on far too many metrics leading to a manuscript that is exceptionally long but without discernible purpose or meaningful conclusions. I recommend that the paper undergo major revisions.

**We appreciate the reviewer's supportive comment on the perturbed parameter model set-up. We believe the manuscript has been significantly improved through revisions based on the reviewer's comments 2.1 to 2.3.**

**To address the reviewer's comment regarding the experiment design and based on the recommendations of the first reviewer (please see comment 1.2.1), we have revised the manuscript to follow the realistic future scenario, SSP1-2.6. Thus, we have redone the analysis to now follow this more standard scenario. However, our essential results have not altered fundamentally.**

**In this study, we use a large ensemble to investigate two climate metrics of the eTCRE and ZEC (the original manuscript, section 1) until the end of the century, which is relevant to the climate policy, and beyond (over several centuries) to understand the long-term implications for climate response after the emissions cease. From the analysis of eTCRE framework, we determined the main contributors of uncertainty over positive, net negative and zero emission phases (original manuscript, section 4.1). The control of the climate response is investigated in two different ways involving (i) the control by the carbon and thermal variables and (ii) the correlation analysis between the model parameters and the slope of change in surface air temperature versus cumulative $CO_2$ emissions. From the analysis of ZEC, we showed the importance of deploying negative emissions technologies for reducing the risk of over-shoot after the emissions cease (original manuscript, section 4.3). We mentioned these points in the original manuscript section 5 and now updated it for the new runs. Additionally, we have made revisions to section 1 to bring more clarity to the manuscript.**

2.2.    General Comments

2.2.1   The authors need to explain where the 82 (86?) variants of GENIE come from. There is a reference to Foley et al., 2016) which presumably first derived the model set-up but a paper of this length should be self-contained and explain to the reader which parameters are being perturbed and how the variants represent uncertainty in the climate system.

**Thank you for raising this point. The 86 variants of GENIE span plausible uncertainty in the physical and biogeochemical components of the model. In the revised manuscript section 3.1, we explained in more detail the characteristics of the 28 parameters and the resulting 86 parameter combinations which used in this study along with the relevant references.**

2.2.2   The experiment design does not make much sense. The design is an example of a half-idealized experiment, as the non-$CO_2$ forcings are frozen at 2020 values but there is no explanation as to why this is done. Why not follow the real RCP 4.5 pathway? There are times when halfidealized experiments do make sense but their purpose needs to be clear which is absent here. Also why use RCP 4.5 instead of SSP 4.5?

**Thank you for pointing these out. We addressed the reviewer's comment by updating the manuscript based on the results of the new simulations for SSP1-2.6 scenario. We chose this scenario over SSP4.5 to investigate the implications of the climate metrics in the best feasible scenario for policy-making purposes, as recommended by the first reviewer (please see comment 1.2.1). The model set-up for this scenario until 2100 and beyond is based on Riahi et al. (2017) and Meinshausen et al. (2020), which allows a continuation of non-$CO_2$ forcings until the end of the run. In the revised manuscript, section 3.1, we detailed the experiment design.**

2.2.3    What is gained by starting the experiments in 850 CE instead of 1850 CE? The relevance model simulations would be easier to interpret if either real scenarios or idealized experiments were conducted.

**In the original manuscript, section 1, we mentioned the reason for choosing 850 CE as the pre-industrial baseline is to account for land use change. This effect of land use change is minimised by starting at 1850 CE. Thus, in this study, the effect of the emissions from both natural and fossil fuel sources are included in the eTCRE calculations. It is notable that the historical transient emissions follow Eby et al. (2013) (the original manuscript section 3.1), in which the emissions from land use change start from 850 CE and the fossil fuel $CO_2$ emissions from 1750 CE (not 1850 CE). In order to bring more clarity to the manuscript, we added the start year for the emissions from each source in the revised version, sections 1 and 3.1.**

**We believe the reviewer's comment has been addressed through the response to comment 2.2.2, as the manuscript has been revised for the more realistic sustainable development scenario, SSP1-2.6.**

2.2.4    Following from the strange experiment design the computation of eTCRE if not comparable to other models and the computation of ZEC is incorrect. eTCRE by its nature varies by the pathway of non-$CO_2$ forcing agents. Here the effect of these agents is frozen in 2020, meaning that eTCRE cannot be compared to eTCRE computed by other models for RCP 4.5.

**Thank you. We believe the reviewer's comment has been addressed through revising the manuscript and redoing all of the model calculations to now be based on a new set of simulations for SSP1-2.6. As mentioned in response to the comment 1.2.1, we applied this scenario until the end of the century and beyond following Riahi et al. (2017) and Meinshausen et al. (2020) in which the non-$CO_2$ radiative forcing evolves in time; thus, the results of the eTCRE can now be compared with the existing literature.**

**We fully addressed the comment regarding ZEC in response to comment 2.2.5.**

2.2.5    ZEC is defined as the Zero Emissions Commitment and can be defined as $CO_2$-only ZEC, or ZEC for other forcing agents independently or in combination (e.g. Matthews & Zickfeld 2012). Here none of these protocols is followed and you attempt to compute ZEC for zero $CO_2$ emissions while holding the non-$CO_2$ forcings constant. Thus the metric you compute is not ZEC, nor can it be compared to ZEC derived from other model simulations.

**In the revised manuscript (section 4.3), we included an additional reference model experiment where the cumulative carbon emissions are kept constant after year 2077, but the non-$CO_2$ forcing is allowed to evolve as in SSP1-2.6. Accordingly, we updated the abstract and section 5.**

2.2.6  While the decomposition of eTCRE into components is fine it is unclear what this analysis adds to our understanding of eTCRE. The paper essentially does the decomposition explains the results but does not assess how the 28 perturbed parameters are controlling the evolution of these components.

**Thank you for raising this point. In the revised manuscript, the abstract and sections 4.2.2 and 5, we explicitly explained how the results of the correlation between the model parameters and the slopes of change in surface air temperature versus cumulative $CO_2$ emissions are linked to the results of the eTCRE framework analysis. During the positive emissions phase, it was shown that the radiative feedback parameter, responsible for more than 81% of the variance in GENIE-1 climate sensitivity (Holden et al., 2010), is the main source of uncertainty in the values of $\Delta T/\Delta I_{em}$. This inference is in agreement with the results derived from analysis of the eTCRE framework, which showed thermal response contribution has a major role in uncertainty in eTCRE through the climate feedback parameter. The results of the two analyses are also consistent for the net negative emission phase. Based on the correlation analysis, the major source of uncertainty is the $CO_2$ fertilisation parameter which drives the main uncertainty in the terrestrial carbon. The uncertainty stemming from the carbon contributions in the eTCRE framework is expressed through the airborne fraction and similarly was diagnosed as the dominant control of uncertainty on eTCRE values during this period. Thus, the two types of analysis provide internally consistent answers.**

2.3. Specific Comments

2.3.1.  Line 2: K EgC^-1 is the preferred unit. Having Celsius and carbon together is confusing.

**We changed the unit °C EgC$^{-1}$ to K EgC$^{-1}$ throughout the revised manuscript, including the abstract, sections 4 and 4.1 and Figs. 6 and S1.**

2.3.2.  Line 180: Does this mean that 4 of the model variants crashed?

**Yes, in the original simulations, 4 out of 86 runs crashed. However, in the new simulations all 86 runs were successfully completed. We omitted the sentence regarding the crashed runs in the revised manuscript, section 3.1.**

2.3.3.  Figure 2: Grey lines for each variant with a mean value would be easier to look at.

**We can see the point that the reviewer is raising, but the reason for choosing different colours in Fig. 2 was to show the behaviour of each individual ensemble member more distinctively, thus, we believe the figure with the current format can serve this purpose better.**

2.3.4.  Figure 3: The release of carbon from sediments seems really high.

**The sedimentary $CaCO_3$ dissolution flux ranges from around 25 to 40 Tmol C yr$^{-1}$ (Archer, 1996; Sulpis et al., 2018) and further up to 43 Tmol C yr$^{-1}$ (Ridgwell and Hargreaves, 2007). During the positive emission phase (until 2077), the carbon release from the sediment reservoir is ~ 14 PgC on average, equivalent to a sedimentary $CaCO_3$ dissolution flux of ~ 21 TmolC yr$^{-1}$ consistent with the existing literature. In the revised manuscript, we added this point to the text in section 3.2.**

2.3.5. Line 224: Does Genie have any other heat sinks? Estimate is wrong word, makes it seem like energy is not being conserved by the model.

**The ocean provides the only heat sink in GENIE-1, aside from melting of ice. However, the reason for using the word estimate in line 224 (section 3.3 in the original manuscript) was solely because the ocean heat uptake was used as an approximate for planetary heat uptake in calculations of thermal terms while it constitutes about 90 % of this value (Church et al., 2011). We mentioned this point in the same sentence in the original manuscript.**

2.3.6. Figure 10: Needs work. Different line styles for the percentiles would improve legibility.

**In the revised manuscript, we assigned different line styles to each $10^{th}$, $50^{th}$ and $90^{th}$ percentile values in Fig. 10.**

**References**

Archer, D.: A data-driven model of the global calcite lysocline, Global Biogeochem. Cy., 10, 511 – 526, doi: 10.1029/96GB01521, 1996.

Church, J. A., White, N. J., Konikow, L. F., Domingues, C. M., Cogley, J. G., and Rignot, E., Gregory, J. M., van den Broeke, M. R., Monaghan, A. J., and Velicogna, I.: Revisiting the Earth's sea-level and energy budgets from 1961 to 2008, Geophys. Res. Lett., 38, L18601, doi:10.1029/2011GL048794, 2011.

Eby, M., Weaver, A. J., Alexander, K., Zickfeld, K., Abe-Ouchi, A., Cimatoribus, A. A., Crespin, E., Drijfhout, S. S., Edwards, N. R., Eliseev, A. V., Feulner, G., Fichefet, T., Forest, C. E., Goosse, H., Holden, P. B., Joos, F., Kawamiya, M., Kicklighter, D., Kienert, H., Matsumoto, K., Mokhov, I. I., Monier, E., Olsen, S. M., Pedersen, J. O. P., Perrette, M., Philippon-Berthier, G., Ridgwell, A., Schlosser, A., Schneider von Deimling, T., Shaffer, G., Smith, R. S., Spahni, R., Sokolov, A. P., Steinacher, M., Tachiiri, K., Tokos, K., Yoshimori, M., Zeng, N., and Zhao, F.: Historical and idealized climate model experiments: an intercomparison of Earth system models of intermediate complexity, Clim. Past, 9, 1111–1140, doi: 10.5194/cp-9-1111-2013, 2013.

Holden, P. B., Edwards, N. R., Oliver, K. I. C., Lenton, T. M., and Wilkinson, R. D.: A probabilistic calibration of climate sensitivity and terrestrial carbon change in GENIE-1, Clim. Dyn., 35, 785–806, doi: 10.1007/s00382-009-0630-8, 2010.

Meinshausen, M., Nicholls, Z. R. J., Lewis, J., Gidden, M. J., Vogel, E., Freund, M., Beyerle, U., Gessner, C., Nauels, A., Bauer, N., Canadell, J. G., Daniel, J. S., John, A., Krummel, P. B., Luderer, G., Meinshausen, N., Montzka, S. A., Rayner, P. J., Reimann, S., Smith, S. J., van den Berg, M., Velders, G. J. M., Vollmer, M. K., and  Wang, R. H. J.: The shared socio-economic pathway (SSP) greenhouse gas concentrations and their extensions to 2500, Geosci. Model Dev., 13, 3571–3605, doi: 10.5194/gmd-13-3571-2020, 2020.

O'Neill, B.C., Kriegler, E., Ebi, K.L., Kemp-Benedict, E., Riahi, K., Rothman, D.S., van Ruijven, B.J., van Vuuren, D.P., Birkmann, J., Kok, K., and Levy, M.: The roads ahead: Narratives for shared socioeconomic pathways describing world futures in the 21st century, Glob. Environ. Change, 42, 169-180, doi: 10.1016/j.gloenvcha.2015.01.004, 2017.

Riahi, K., van Vuuren, D. P., Kriegler, E., Edmonds, J., O'Neill, B. C., Fujimori, S., Bauer, N., Calvin, K., Dellink, R., Fricko, O., Lutz, W., Popp, A., Cuaresma, J. C., Samir, K.C., Leimbach, M., Jiang, L., Kram, T., Rao, S., Emmerling, J., Ebi, K., Hasegawa, T., Havlík, P., Humpenöder, F., Da Silva, L. A., Smith, S., Stehfest, E., Bosetti, V., Eom, J., Gernaat, D., Masui, T., Rogelj, J., Strefler, J., Drouet, L., Krey, V., Luderer, G., Harmsen, M., Takahashi, K., Baumstark, L., Doelman, J. C., Kainuma, M., Klimont, Z., Marangoni, G., Lotze-Campen, H., Obersteiner, M., Tabeau, A., and Tavoni. M.: The Shared Socioeconomic Pathways and their energy, land use, and greenhouse gas emissions implications: An overview, Glob. Environ. Change, 42, 153-168, doi: 110.1016/j.gloenvcha.2016.05.009, 2017.

Ridgwell, A., and Hargreaves, J. C.: Regulation of atmospheric $CO_2$ by deep-sea sediments in an Earth system model, Global Biogeochem. Cy., 21, doi: 10.1029/2006GB002764, 2007.

Sulpis, O., Boudreau, B. P., Mucci, A., Jenkins, C., Trossman, D. S., Arbic, B. K., and Key, R. M.: Current $CaCO_3$ dissolution at the seafloor caused by anthropogenic $CO_2$, Proc. Natl. Acad. Sci. U. S. A., 115, 11700-11705, doi: 10.1073/pnas.1804250115, 2018.

Vakilifard, N., Kantzas, E. P., Holden, P. B., Edwards, N. R., and Beerling, D.J.: The role of enhanced rock weathering deployment with agriculture in limiting future warming and protecting coral reefs, Environ. Res. Lett., 16, 094005, doi: 10.1088/1748-9326/ac1818, 2021.

---

## Author Response (AR2)

Dear Dr. Bond-Lamberty,
We are grateful for the positive feedback on the revised manuscript and thank the reviewer for their time and constructive feedback. We provided a detailed response to the reviewer's comments and revised the manuscript for further improvement.

Sincerely,
Negar Vakilifard, Corresponding author

*Detailed response to the reviewers' comments*

Overall Assessment

The revised version of the paper is greatly improved from the original version and most of my critiques have been satisfactorily addressed. There are a few remaining issues that need to be addressed before publication but these can be addressed with more careful explanations and wording of the paper. Once these issues are addressed I believe that the paper will be ready for publication.

**We thank the reviewer for their supportive comments on the revisions made to address the critiques. We have implemented the reviewer's suggestions in this version of the manuscript to improve the clarity of the paper.**

1. General Comments

1.1. While improved from the original version of the paper the use of ZEC is still not quite correct. ZEC is 'Zero Emission Commitment', however in your experiments emissions of non-$CO_2$ greenhouse gases and aerosols continue for the period where ZEC is assessed. To avoid confusions and apples to oranges comparisons I recommend defining an 'effective Zero Emissions Commitment' eZEC analogous to eTRCE used in the paper. By clearly defining such a metric future research and reviews will not so easily get tripped up by inconsistent definitions of ZEC.

**Thank you for your insightful comment. We agree with the recommendation in using an eZEC notation that parallels an eTCRE notation when non-$CO_2$ forcing is included. In the revised manuscript, section 1, we have changed to the recommended notation of the effective ZEC referring to the continued surface warming after the cessation of the $CO_2$ emissions while the non-$CO_2$ greenhouse gases and aerosol forcings evolve. Accordingly, we revised Figure 11, sections 5, 5.3, 6 and the abstract.**

1.2 Organization of the paper is a bit weird. Results basically begin at line 200, part-way through the Methods section.

**Thanks for your comment. We removed the parts related to the results from section 3.1 in the previous version and moved them along with sections 3.2 and 3.3 to a new section (section 4) with the heading "GENIE-1 model responses". We then re-numbered the following headings.**

2. Specific Comments

2.1. Abstract: There is not enough context in the abstract to clearly understand what Lines 41 to 42 and 45 to 46 mean. I recommend re-writing the abstract to either add context or remove these sentences.

**Thank you for bringing this up. We addressed this comment by removing line 41 from the abstract and revising lines 45 and 46. It now reads: "If net negative emissions are included, there is a reduction in atmospheric $CO_2$ and there is a decrease in temperature overshoot, so that the eZEC is positive in only 5 % of the ensemble members".**

2.2. Line 53: Add a citation to the text of the Paris Agreement.

**In the revised manuscript, we added the reference for the Paris Agreement in sections 1 and references.**

2.3. Line 87: Add a comma after 'biosphere'

**We made this revision in the manuscript, section 1.**

2.4. Line 100: 86 is not really a 'large' ensemble. Similar studies have used 250 or 1000 model variants (e.g. Steinacher & Joos 2016, MacDougall et al. 2017). Maybe just say 'an ensemble'

**Thank you for pointing this out. We have replaced the term 'large ensemble' with 'an ensemble' throughout the revised manuscript, including the title of the paper. Please also see sections 1 and 6.**

2.5. Line 101: You should add a sentence to acknowledge the very high uncertainty in land-use reconstructions prior to about 1800 CE. Although the 850 CE start date was used by Eby et al. 2013 and other models intercomparison and studies, it is now well known that the land-use reconstructions used for forcing those experiments was based a very poor population estimate data-sets and Eurocentric conceptualizations of land-use. See Koch et al 2019, for a review of the problems in the Americas (the reconstruction used by Eby et al is designated P08 in Koch et al 2019).

**Thank you. We have addressed this comment by adding a sentence in section 1, mentioning the limitations of the land use change emission reconstructions prior to 1800 CE and cited the recommended reference.**

2.6. Line 103: Add 'climate' after pre-industrial

**We made this revision in the manuscript, section 1.**

2.7. Line 181: How did you account for the transition from historical RCP datasets ending in 2005 to SSP 2.6 beginning in 2015? Are there any discontinuities in the forcing data-sets?

**Thank you for your comment. There are no discontinuities in the forcing data-sets. We made an adjustment to the non-$CO_2$ radiative forcing of SSP1-2.6 by adding a constant value of 0.446 $Wm^{-2}$ to make it consistent with the RCP2.6 spin-up at 2005. This adjustment can be reconciled as contributions from land-use change albedo (explicitly modelled in GENIE-1, Fig 1c) and from non-anthropogenic forcings which were modelled in the historical spin-up (Eby et al., 2013), comprising volcanic forcing of 0.184 $Wm^{-2}$, and solar forcing of 0.059 $Wm^{-2}$ in 2005. In the revised manuscript, we added this explanation to section 3.**

2.8. Figure 1: Add CE after year for clarity. Model years and years BP are also commonly used in ESM studies.

**In the revised manuscript, we added CE after year in Fig. 1.**

2.9. Figure 2: Gray lines with a mean value in black or another colour on-top would be clearer than the rainbow shown.

**We revised Fig. 2 and showed the spread of the results in grey and the mean values in black.**

2.10. Line 254 and 255: You are conflating the natural world and your model here. In your model the ocean is the only major energy sink, while it is in the natural world that the ocean takes up ~90% of heat.

**Thank you for bringing this up. In the revised manuscript, section 4.3, we mentioned that the ocean heat uptake is used to represent the planetary heat uptake, as the model ocean is the principal energy sink and the model does not take into account the energy stored in the lithosphere or consumed in the melting of the ice sheets. We then added that in real world, the ocean is responsible for storing over 90 % of the Earth's total energy increase (Church et al., 2011).**

2.11. Line 273: Also cite Koven et al. 2022

**We added this reference in the revised manuscript, sections 5 and references.**

2.12. Figure 7: Be clear that these are cumulative not instantaneous fractions.

**Thank you for your comment. In the caption of Fig. 7, we added that the y-axis shows the cumulative fraction of $CO_2$ which remains in each carbon inventory.**

2.13. Line 425: Cite MacDougall et al. 2017 here, which had a similar result for climate sensitivity.

**We cited the recommended reference in the revised manuscript, section 5.2.2.**

2.14. Line 476: Reword for clarity. When a nation-state no longer wants to abide by a treaty they 'leave' it. Thus the wording here is very confusing. Revise to specifically mention temperature targets.

**Thank you. In the revised manuscript, section 6, we added the Paris agreement temperature targets to bring more clarity to the manuscript.**

*References*

Church, J. A., White, N. J., Konikow, L. F., Domingues, C. M., Cogley, J. G., and Rignot, E., Gregory, J. M., van den Broeke, M. R., Monaghan, A. J., and Velicogna, I.: Revisiting the Earth's sea-level and energy budgets from 1961 to 2008, Geophys. Res. Lett., 38, L18601, doi:10.1029/2011GL048794, 2011.

Eby, M., Weaver, A. J., Alexander, K., Zickfeld, K., Abe-Ouchi, A., Cimatoribus, A. A., Crespin, E., Drijfhout, S. S., Edwards, N. R., Eliseev, A. V., Feulner, G., Fichefet, T., Forest, C. E., Goosse, H., Holden, P. B., Joos, F., Kawamiya, M., Kicklighter, D., Kienert, H., Matsumoto, K., Mokhov, I. I., Monier, E., Olsen, S. M., Pedersen, J. O. P., Perrette, M., Philippon-Berthier, G., Ridgwell, A., Schlosser, A., Schneider von Deimling, T., Shaffer, G., Smith, R. S., Spahni, R., Sokolov, A. P., Steinacher, M., Tachiiri, K., Tokos, K., Yoshimori, M., Zeng, N., and Zhao, F.: Historical and idealized climate model experiments: an intercomparison of Earth system models of intermediate complexity, Clim. Past, 9, 1111–1140, doi: 10.5194/cp-9-1111-2013, 2013.

Koch, A., Brierley, C., Maslin, M. M., and Lewis, S. L.: Earth system impacts of the European arrival and Great Dying in the Americas after 1492. Quat. Sci. Rev., 207,13-36, doi: 10.1016/j.quascirev.2018.12.004, 2019.

Koven, C. D., Arora, V. K., Cadule, P., Fisher, R. A., Jones, C. D., Lawrence, D. M., Lewis, J., Lindsay, K., Mathesius S., Meinshausen, M., Mills, M., Nicholls, Z., Sanderson, B. M., Séférian, R., Swart, N. C., Wieder, W. R., and Zickfeld, K.: Multi-century dynamics of the climate and carbon cycle under both high and net negative emissions scenarios, Earth Syst. Dyn., 13, 885-909, doi: 10.5194/esd-13-885-2022, 2022.

MacDougall, A. H., Swart, N. C., and Knutti, R.: The uncertainty in the transient climate response to cumulative $CO_2$ emissions arising from the uncertainty in physical climate parameters, J. Clim., 30, 813-27, doi: 10.1175/JCLI-D-16-0205.1, 2017.

Steinacher, M., and Joos, F.: Transient Earth system responses to cumulative carbon dioxide emissions: linearities, uncertainties, and probabilities in an observation-constrained model ensemble, Biogeosciences, 13, 1071-1103, doi: 10.5194/bg-13-1071-2016, 2016.